# The Shirom-Melamed Vigor Measure for Students: Factorial Analysis and Construct Validity in Spanish Undergraduate University Students

**DOI:** 10.3390/ijerph17249590

**Published:** 2020-12-21

**Authors:** Manuel Pulido-Martos, Daniel Cortés-Denia, Juan José de la Rosa-Blanca, Esther Lopez-Zafra

**Affiliations:** School of Humanities and Sciences of Education, Department of Psychology, University of Jaén, Campus Las Lagunillas s/n, 23071 Jaén, Spain; dcortes@ujaen.es (D.C.-D.); juanjodelarosablanca@gmail.com (J.J.d.l.R.-B.); elopez@ujaen.es (E.L.-Z.)

**Keywords:** academic performance, mental health, physical activity, satisfaction with life, vigor

## Abstract

Students suffer from a decrease in physical activity during their education period. This lower level of activity could affect, through various paths, their academic performance, mental health, and satisfaction with life. In these two studies, we assumed that vigor, a positive affect variable, would act as a mediating variable in the above relationship, and thus, we proposed an instrument for evaluating vigor in academic contexts. In Study 1, 707 undergraduates (59.7% women) responded to the vigor scale adapted for students to test factorial validation (through confirmatory factor analysis) and obtain reliability indicators. In Study 2, 309 undergraduates (55.3% women) completed a questionnaire measuring physical activity, mental health, satisfaction with life, vigor, and academic performance to test a structural model of the relationships between the variables to obtain construct validity. A measurement model with three related factors, each representing one dimension of vigor, optimally fit the data, and the reliability indices were adequate (Study 1). Moreover, the mediational model confirmed a complete influence of physical activity on satisfaction with life, academic performance, and mental health levels through students’ vigor levels with optimal adjusting values (Study 2). Proposing an instrument such as the Shirom-Melamed Vigor Measure for students allows the opening of a research venue that is focused on the study of positive affects in academic contexts, as well as the testing of the physical activity pathways of action in obtaining positive results.

## 1. Introduction

In addition to the beneficial effects on physical health [1,2,3], performing physical activity (PA) in adulthood is related to a reduced risk of mental ill-health [4] and has been shown to be effective in reducing symptoms associated with depression [5,6]. Going beyond a clinical approach and considering mental health (MH) as a positive state of well-being related to an optimal psychological experience and functioning [7], there are several studies that find a positive relationship between PA and satisfaction with life (SWL) [8,9,10], i.e., the extent to which people value their present situation as an ideal situation [11]. Cognitive function is also positively affected by performing PA [12,13], and positive relationships have been found between PA and academic performance (AP) in academic contexts, such as the university [14,15].

Despite the relationships found between PA and different outcomes in MH, SWL, and AP, the underlying mechanisms of this association are not fully understood [16,17,18]. In this vein, conceptual models such as that of Lubans et al. [19] are of great interest in identifying the neurobiological, psychosocial, and behavioral mechanisms that mediate the effects of PA on outcomes such as cognitive function, well-being, and ill-being. There is a rooted tradition in the study of biological and/or neuronal factors that intervene in the positive effects derived from performing PA [20] compared to the inclusion of emotional or affective factors, even though affect has shown its usefulness in the explanation of results in health, educational, and work contexts, among others [21,22,23]. In the workplace, the positive and energetic affect known as vigor [24,25], which is defined as “a positive affective response to one’s ongoing interactions with significant elements in one’s job and work environment that comprises the interconnected feelings of physical strength, emotional energy, and cognitive liveliness” (p. 143) [24], has turned out to be a mediating variable of interest in the relationship that a group of personal and work resources [26] have with work satisfaction [27,28,29], MH [30], and work performance [28,31,32], among others. Academic work and the context in which it is performed show similarities with work activity and the framework in which it occurs, i.e., the organization [33]. In fact, constructs whose origins are situated in an employment context (burnout, engagement, organizational justice, etc.) have been successfully used in academic contexts [34,35]. Considering vigor at work [24] in academic contexts could help to understand the psychosocial-type mechanisms by which PA exerts its effect on cognitive function, health, and well-being. The theoretical bases that support the mediating role of affective experiences in the achievement of results can be found in theories such as the conservation of resources [36] or the broaden-and-build model of positive affects [37]. PA can be seen as an internal resource in that it helps people achieve their goals [38] and the exposure to stable resources (internal or external) generates affective experiences, including vigor [39]. The accumulation of resources, along with students feeling strong on a physical, cognitive, and emotional level, allows students to experience more vigor, which favors the achievement of positive results [25,37]. The mediating role of vigor has not been analyzed in educational contexts but has received empirical support in work contexts [31,32]. In educational contexts, the analysis of personal resources of a cognitive nature and related to a function of resilience has predominated [40,41]. Thus, we consider that a resource such as vigor at work, which in addition to a cognitive dimension includes affective experiences referring to the social and physical aspects of the person, can be a useful and complete resource in its application to an academic context.

Currently, there is no measure in academic contexts that allows for the evaluation of vigor from the perspective of Shirom [24,25]. The Shirom-Melamed Vigor Measure (SMVM) [24] assesses vigor at work, but it is designed for workers. The SMVM has recently been adapted to different countries [29,42] but its psychometric properties have not been analyzed in academic contexts.

In sum, the aim of this paper was twofold: 1) to determine the factorial structure and reliability of the SMVM in students, i.e., the SMVM for students (SMVM-S) (Study 1), and, thus, education professionals may have a new instrument with adequate psychometric properties to evaluate the affective experiences of their students; and 2) to analyze the construct validity of the SMVM-S by testing a structural model of the relations among PA, AP, and MH by including vigor as a mediator (Study 2), which could help testing the role of vigor in the explanation of different useful and highly valued results in educational contexts.

## 2. Study 1: Validation of the Factorial Structure of SMVM-S

This study was performed to analyze the factor structure, validity, and internal consistency of the SMVM in the university context.

### 2.1. Materials and Methods

#### 2.1.1. Design, Participants and Procedure

A cross-sectional survey was performed and completed by undergraduates from several degree programs at different Spanish universities. Participation was voluntary, and informed consent was obtained. All the procedures followed the protocol approved by the Ethics Committee of the University of Jaén (code: CEIH 171215). The participants were recruited by 14 students in their last year from the Psychology grade and the Official Master’s Program of Positive Psychology at the University of Jaén (Spain), who were instructed about the procedure and the distribution of the questionnaires. In total, 707 participants (out of 709 initial; 2 were eliminated due to incomplete data) completed the SMVM-S questionnaire and the other scales during the second semester of the academic year. The participants’ mean age was 21.1 years (SD = 2.9; range 17 to 48 years) and 59.7% of them were female. The distribution by areas was: 67.5% from Social and Legal Sciences, 13.6% from Engineering and Architecture, 8.3% from Arts and Humanities, 5.9% from Sciences and 3.3% from Health Sciences, the remaining 1.4% were missing values. 

#### 2.1.2. Measures

Demographics: The students reported their sex, gender, grade they were enrolled in, and their university of origin.

Vigor: Stemming from the Shirom-Melamed Vigor Measure (SMVM) [24] and its Spanish adaptation [42], items were modified by substituting the content referring to employees and/or clients with the term “class mates”. The participants had to indicate how often in the past 30 days they had felt a number of different feelings while studying. The scale consisted of 12 items that comprised three dimensions: physical strength (5 items, i.e., “I feel full of energy”), cognitive liveliness (3 items, i.e., “I feel I can think rapidly”), and emotional energy (4 items, i.e., “I feel able to show warmth to others”). The response format ranged from 1 (almost never) to 7 (almost always). The scale yielded an adequate reliability, with ρ values of 0.87, 0.73, and 0.83 [42].

#### 2.1.3. Data Analysis

Confirmatory factor analysis (CFA) determined the factorial structure and the goodness of fit of the different models of measurement. The violation of the assumption of multivariate normality of the data (Mardia’s coefficient = 36.79) suggested the use of a robust method of maximum likelihood (ML Robust) with EQS 6.4 software (Multivariate Software Inc., Temple, CA, USA). As indicators of the goodness of fit of the measurement models to the data, the corrected χ^2^ by Satorra–Bentler (S-Bχ^2^), NNFI (Non-normed fit index), CFI (Comparative fit index), SRMR (Standardized root mean square residual), and RMSEA (Root mean square error of approximation), as well as their 90% confidence intervals (90% CI) were used. An acceptable model fit was defined as NNFI and CFI values close to 0.95 or greater, a SRMR value close to 0.08 or below, and a RMSEA value less than 0.08 [43,44]. The reliability of the dimensions of the instrument were examined by ρ [45], with 0.70 as an acceptable cut-off value [46]. The composite reliability (CR) and average extracted variance (AVE) were analyzed as indicators of internal consistency and convergent validity of the variables included, respectively. Cutoff values considered acceptable were CR values over 0.70 and AVE values over 0.50 [47]. Moreover, 0.6% of the values for all the variables were missing. Missing data were imputed using the expectation maximization algorithm in IBM SPSS (IBM Corp., Armonk, NY, USA) for Windows v22.0 after checking that the missing values were completely random using Little’s MCAR (missing completely at random) test [48] (χ^2^ = 22.67; df = 43; *p* = 0.995).

### 2.2. Results

Table 1 shows the resulting adjustment values of the CFA for different measurement models. The measurement models proposed in the adaptation to Spanish of the SMVM [42] were tested, along with other models that have been confirmed in workers. Specifically, the other models included a model consisting of a single-factor structure (M1), a hierarchical model with three first-order factors representing the dimensions of vigor and a second-order factor (M2), a model with three related factors, each representing one dimension of different vigor (M3), a model equal to M3 but allowing the correlation of some error terms (M4), and a bifactor model (M5). The results show inadequate rates for M1 and M5. Models 2, 3, and 4 are nested models; thus, the differences in S-Bχ^2^ (Satorra–Bentler chi square) when compared to each other show the best fit for M4. This model, proposed from the revision of the modification indices (MI), suggests the inclusion of a correlation between the error terms of items 4 and 5 of the SMVM-S.

Taking M4 as the model with a better fit to the data (S-Bχ^2^ = 231.54; df = 50; *p* < 0.001; CFI = 0.95; NNFI = 0.93; SRMR = 0.07; RMSEA = 0.07 (0.06–0.08)), in Table 2, the standardized and non-standardized coefficients of the measurement model are shown, as well as the indices related to reliability. All the standardized factor loadings are significant at *p* < 0.001 and range from 0.61 to 0.96. Correlations between SMVM-S factors are all significant, with coefficients of 0.43 (physical strength-cognitive liveliness), 0.37 (physical strength-emotional energy), and 0.37 (cognitive liveliness-emotional energy). The indicators related to the reliability of the factors showed acceptable values in all cases.

## 3. Study 2: Construct Validation

In this study, the validity of the construct was assessed by testing a structural model that analyzed the joint role of PA and vigor in undergraduates’ MH, SWL, and AP.

### 3.1. Materials and Methods

#### 3.1.1. Design, Participants and Procedure

Data were obtained from a cross-sectional survey completed by undergraduates from several Spanish universities who were studying in different degree programs. Participation was voluntary, and the students provided informed consent. All procedures followed the research protocol approved by the Ethics Committee of the University of Jaén (code: CEIH 171215). The participants were recruited over three months (February to April) by four Master’s students (Official Master’s Program in Positive Psychology and Official Master’s Program of Research and Teaching in Physical Activity and Health Sciences from the University of Jaén) who were instructed about the procedure and distribution of the questionnaires.

In total, 309 participants completed this study (out of the initial 312; 3 were eliminated due to incomplete questionnaires). The participants’ mean age was 22.1 years (SD = 2.4; range 18 to 33 years), and 55.3% of them were female. According to the distribution of areas of knowledge, in Spain, 53.4% were ascribed to Social and Legal Sciences, 20.7% to Engineering and Architecture, 9.1% to Arts and Humanities, 2.3% to Sciences, and 14.2% to Health Sciences; the remaining 0.3% were missing values.

#### 3.1.2. Measures

Demographics: The same data as those used in Study 1 were included.

Vigor: The same as that described in Study 1.

Physical activity: The International Physical Activity Questionnaire-Short Form (IPAQ-SF) [49] was administered. This instrument allows for the calculation of the volume of total PA (walking: moderate and vigorous) over seven days by evaluating or assigning to each activity some energy requirements, which are defined as METs (multiples of the rate of metabolic expenditure). This allowed us to obtain a result in METs-minutes. This entails multiplying the result of the METs of an activity by the number of minutes during which it has been performed. A total MET-minutes per week number was obtained for each participant, where the highest values indicate a higher volume of total PA per week and, therefore, a higher energy consumption. The psychometric properties of the instrument have been evaluated in different countries and have obtained optimal values of reliability and validity in the adult population [49,50]. In Spanish undergraduates, the IPAQ-SF was shown to be a valid instrument when compared to objective measurements of PA obtained using accelerometers [51].

Mental health: The General Health Questionnaire (GHQ) [52] assesses normal healthy functioning to measure minor psychiatric morbidity and the appearance of new, distressing symptoms [53]. This study includes the 12 items version (GHQ-12) [54] adapted to Spanish [55]. This scale includes items such as “Could you concentrate as well as you used to?” that participants responded to in a 4-point Likert format (ranging from 0 = much more than usual to 3 = not at all). Despite some controversy regarding the factorial structure of the instrument, studies with the Spanish population confirm a structure of three interrelated factors [56]: dysphoria, social dysfunction, and loss of confidence. Other studies also confirmed these factors [57,58]. A higher score indicates lower levels of health. In general, the Cronbach’s alpha for Spanish samples is 0.76 [55].

Satisfaction with life: The Spanish version [59] of the Satisfaction with Life Scale (SWLS) [60] was included in the questionnaire. This instrument has an unidimensional structure and assesses the cognitive component of subjective well-being [61]. On a 5-point Likert scale (ranging from 1 = not at all to 5 = absolutely), the participants responded to 5 items (i.e., “In most ways, my life is how I want it to be”). In the Spanish general population, the Cronbach’s alpha value is 0.88 [59].

Academic performance. To evaluate AP, the students’ grades were considered. The overall grade was determined as the weighted average of the final scores obtained in all subjects, with a range from 0 to 10, where 10 is the highest grade. Freshman students indicated their global score in access to university exams, which was recorded on a 14- to 10-point scale. In our sample, the range was from 5 to 9.21 points, with a mean grade of 6.98 (SD = 0.93).

#### 3.1.3. Data Analysis

All analyses were performed with EQS 6.4 software for Windows (Multivariate Software Inc., Temple City, CA, USA). To avoid model estimation problems due to the disparate metrics of the variables, the items were rescaled by means of the percentage of proportion of maximum scoring (POMS) [62] using the observed maximum score in each variable. Due to the deviation of the data regarding the multivariate normality (Mardia’s coefficient = 10.29), the option ML Robust was used. The fit of the measurement model to the data was first verified by CFA. The model was shaped by five latent variables: vigor with three indicators (physical strength, cognitive liveliness and emotional energy), total PA with an indicator (MET-minutes per week), SWL with five indicators (each item of the scale), MH with three indicators (dysphoria, social dysfunction, and loss of confidence), and AP with one indicator (average grade taken from the student record). For latent variables with a single indicator, the factor loadings were set with a value equal to one, and the variance of the error was estimated to fix it a priori [63]. Finally, a covariation was allowed between the error terms of two indicators of MH (dysphoria and loss of confidence) that would be theoretically justified [64]. The indicators of the goodness of fit of the models, as well as the criteria for deciding on their fit, were the same as those used in Study 1. Regarding the analysis of the structural model, first Pearson’s correlation coefficients were calculated for the relationships between the variables analyzed and, then, following the theoretical mediational model of vigor at work [25] and theoretical proposals as Lubans’ et al. [19], the fit of a group of non-nested structural models based on the CAIC (Consistent Akaike information criterion) index was compared. Lower values of the index corresponded to a better fit of the model, as long as the rest of the indices and their values met the decision criteria used in Study 1. Moreover, less than 3% of the values for all the variables on each scale were missing. Following the same procedure as that used for the missing data from Study 1, the missing values were imputed using the expectation maximization algorithm, since the missing values were completely random, and by using Little’s MCAR test, para SMVM-S (χ^2^ = 47.54; df = 42; *p* = 0.257), GHQ (χ^2^ = 72.34; df = 66; *p* = 0.277), SWLS (χ^2^ = 9.85; df = 12; *p* = 0.629), and the average grade (χ^2^ = 5.78; df = 7; *p* = 0.565).

### 3.2. Results

The adjustment values of the measurement model tested were adequate (S-Bχ^2^ = 93.87; df = 56; *p* < 0.01; CFI = 0.93; NNFI = 0.91; SRMR = 0.04; RMSEA = 0.05 (0.03–0.06)), resulting in all significant factor loadings (values between 0.39 and 0.92).

Table 3 includes the Pearson correlation coefficients for the relationships between the entire set of variables. Regarding the structural models, the M1 model or the saturated model was compared with the M2 (no direct effect), M3 (no effect from the predictor variable to the mediator), and M4 (no effect from the mediator to the outcome) models [65]. Table 4 shows the adjustment indices for each of the structural models, allowing the CAIC index, since they are non-nested models, and shows that M2 is the best model (CAIC = −308.643), i.e., a model of total mediation in which the levels of PA exert their influence on academic results and levels of SWL and MH through the levels of vigor of the students. The fit indices of the M2 model show adequate values (S-Bχ^2^ = 108.82; df = 62; *p* < 0.001; CFI = 0.92; NNFI = 0.90; SRMR = 0.05; RMSEA = 0.05 (0.03–0.06)). Figure 1 graphically displays the mediation model with the standardized regression coefficients and the percentage of variance explained in each endogenous variable. Total weekly PA levels were positively associated with student vigor levels, and vigor was positively related to SWL levels and AP and negatively related to the levels of MH (note that a higher score in MH implies a more deteriorated state of MH). The model explained 4% of the variance in the levels of vigor, 54% of the levels of SWL, 71% of the levels of MH, and 6% of the AP. The indirect effects that total PA exerted through vigor on levels of SWL, levels of MH and AP were all significant (β = −0.14, β = −0.16, and β = 0.04, respectively), thereby confirming the important mediating role played by the students’ level of vigor in the relationship between the levels of total PA and other important results in academic contexts [66].

## 4. Discussion

Vigor at work has emerged as a positive affect variable, characterized by showing physical strength, cognitive liveliness, and emotional energy [24]. This helps to explain positive consequences at work [25]. In this research, we performed two studies to analyze the psychometric properties of the SMVM-S, an instrument to assess vigor levels in the academic context. Study 1 analyzed the fit to the data of different measurement models of the SMVM-S, as well as the reliability of the instrument. This study confirmed that the model with the best fit consisted of three interrelated factors, following the same structure as that used in previous studies carried out on workers [27,29,42], with significant correlations among the SMVM-S dimensions and adequate indices of internal consistency. Study 2 confirmed the mediational model proposed by Shirom [25] in an academic context and contributed to the identification of affective factors that explain the effects of PA on different results [19]. The level of total PA per week would imply higher levels of vigor in students, while vigor would positively influence academic results and SWL levels and negatively influence the deterioration of MH. With few exceptions [67], the PA-vigor positive relationship (mainly the physical aspect of vigor) has been verified in different studies [68,69,70,71,72]. Vigor in university students derived from performing PA was characterized by a combination of moderate amounts of arousal and pleasure [73,74], affective experiences that have also been associated with physical exercise [75]. In the same way, it has been possible to verify that the levels of vigor in students have an effect on their average grades—as an indicator of performance on their MH—with a lower incidence of psychological problems, and on their SWL. These results contribute to understanding psychosocial mechanisms, specifically the experimentation of the positive affect, by which PA affected the well-being and cognitive performance of university students. Therefore, these results respond to the need to verify the mediational role of constructs such as the vigor, which is proposed in different heuristic and conceptual models [19,25]. In this type of model, it is clear that vigor does not only depend on internal resources, such as PA. External resources related to organizational, group, and task factors (i.e., those related to the design of the educational system) would also be responsible for the levels of affect experienced by the students. This could explain the low percentage of vigor level variance. Obviously, in future studies, the variables related to the design of the educational system should be considered.

In a similar vein, when we observed the values of the explained percentage of PA variance, MH and SWL, we could consider that the combination of higher levels of PA and academic vigor had a greater impact on the processes of affecting health and well-being than on motivational processes related to AP.

However, the psychological correlates of the average grade of the academic record varied, including the intellective aspects, motivational and self-regulatory process, and personality traits or environmental factors [76]. In our study, we did not consider such variables, so this aspect could explain the modest percentage of variance obtained in AP against other variables.

Providing to education professionals an instrument such as the SMVM-S would help to better understand the role that PA has in reducing risks and improving the well-being of university students, who show a high rate of psychological disorders [77], and a decrease in AP compared with other educational stages [78] and whose age includes the adult population group with lower levels of SWL (18–25 years) [79]. In fact, if we consider that a high percentage of university students (40–70%) can be considered physically inactive [80,81,82,83], then the potential for PA to generate positive effects through the levels of vigor experienced is greater [84]. Our results also help to clarify the role of an affective type of variable such as vigor in one stage, namely, the university, which entails important changes in different facets [71,72] for young people. The developing of programs, under the paradigm of positive psychology, to improve the affective experiences of students [85], could be used in university contexts to improve the vigor levels of students. In addition, it could contribute to develop the "whole student" [86]. As part of these interventions, it would be useful to assess other affective motivational variables, such as the levels of student engagement that prove their relationship with the levels of academic vigor levels. Thus, it could be determined whether, as already happens with other positive emotions [87], vigor manages to explain the level of student engagement. Another practical implication of these study results pertains to university heads having to use an instrument such as SMVM-S in order to evaluate the impact of the design of the educational system as a whole, on the levels of affect, specifically in vigor, or the introduction of new guidelines or changes that could affect organizational, group, or task factors. The results derived from the evaluation of the affective experiences of students could help address the influence of these factors on the levels of MH, SWL, and AP.

This study is not absent of limitations. The cross-sectional design prevented us from asserting the order of the relations yielded in Study 2’s mediation model. In future research, designs allowing the evaluation of the analyzed variables at different time points during an academic year, could help to unveil the meaning of the relationships and would also allow researchers to test the fit of more complex sequential-type mediation models. The current study considered total PA without differentiating between light, moderate, or vigorous PA or the different effects that these activity levels could have on the results. To make this decision, we used the potential benefits of PA even when the minimum levels recommended by international organizations were not reached [1,2,3], and the positive effect of PA on AP, regardless of the type of PA [88]. However, we are aware that, in future studies, the mediating role of vigor should be verified by taking into account the different levels of PA and other variables such as environmental aspects [89]. Using self-reporting measures for the assessment of PA could be considered another limitation. Although this instrument concurs with other objective forms of evaluation, such as accelerometry [51], it is recommended that, in future studies, the relationships of the model with other types of measurement be verified. An additional limitation has to do with the sample characteristics in Study 2. Part of this sample (14.6%) were freshmen and their grade for AP was based on the Spanish University Access Tests, a non-compulsory exam taken by students after secondary school, necessary to get into University. Thus, this could undermine the explanation of levels of PA during the university course as being indirectly responsible for a grade obtained previously. To ensure that this did not compromise our results, we performed sensitivity analysis, excluding freshmen students, and the results did not change. 

Finally, all the data were collected in between exam periods, whereas difficulties in managing exercise are greater during examination periods [90]. Thus, future studies should include different temporal moments to determine whether exam periods or other stressful situations influence the relations shown in this study.

## 5. Conclusions

The analysis of the vigor approach [25] in a university context allowed us to test one of the mechanisms proposed at the psychosocial level by Lubans et al. [19], whereby PA affects the AP, MH and SWL of students through the effect of PA on vigor. In achieving the first objective of our study, we provided education professionals with a new evaluation instrument that had adequate psychometric properties, which allowed them to evaluate the affective experiences of their students in relation to their work environments. Regarding the second objective, our results clarified the role of vigor in explaining different useful and highly valued results in educational contexts. Thus, this study offers education professionals an instrument (i.e., the SMVM-S) to make progress in improving the MH, SWL, and AP of university students through the promotion of PA. For those responsible for educational institutions, having this kind of instrument makes it possible to address the impact that factors related to the organization of the educational system have on the affective experiences of students. Currently, and due to the COVID-19 pandemic, it is of special interest to analyze if the forced reorganization faced by the educational centers is causing changes in the levels of academic vigor experienced by the students. Moreover, we could analyze if lockdowns that favor a sedentary lifestyle also promoted a reduction in levels of positive affect [91], and if these effects have the same impact on levels of vigor.

## Figures and Tables

**Figure 1 ijerph-17-09590-f001:**
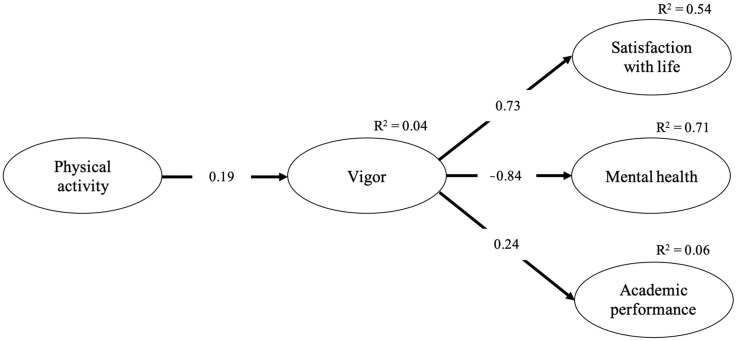
Estimated standardized regression and squared multiple regression (R^2^) coefficients for Model 2 (no direct effect). Note. All estimated standardized regression and squared multiple regression (R^2^) coefficients were significant.

**Table 1 ijerph-17-09590-t001:** Goodness of fit indices for the tested models.

	χ^2^ S-B ^a^	df ^b^	NNFI ^c^	CFI ^d^	SRMR ^e^	RMSEA ^f^	90% CI ^g^ of RMSEA	Δ χ^2^ S-B ^h^
Model 1	1239.58 ***	54	0.57	0.65	0.15	0.18	(0.17–0.19)	-
Model 2	300.81 ***	52	0.91	0.93	0.09	0.08	(0.07–0.09)	-
Model 3	275.64 ***	51	0.91	0.93	0.07	0.08	(0.07–0.09)	40.37 ***
Model 4	231.54 ***	50	0.93	0.95	0.07	0.07	(0.06–0.08)	33.94 ***
Model 5	604.61 ***	42	0.74	0.83	0.08	0.14	(0.13–0.15)	-

^a^ Satorra–Bentler chi square. ^b^ Degrees of freedom. ^c^ Non-normed fit index. ^d^ Comparative fit index. ^e^ Standardized root mean square residual. ^f^ Root mean square error of approximation. ^g^ Confidence interval. ^h^ Scaled Satorra–Bentler chi square difference. *** *p* < 0.001.

**Table 2 ijerph-17-09590-t002:** Standardized and unstandardized coefficients for CFA Model 4 and reliability indicators.

Items	*B* ^a^	SE ^b^	β ^c^	R^2 d^	Factor	ρ ^e^	CR ^f^	AVE ^g^
SMVM-S 1	1.00	0.00	0.79	0.62	Physical strength	0.84	0.91	0.66
SMVM-S 2	1.15	0.05	0.84	0.71				
SMVM-S 3	1.09	0.05	0.81	0.65				
SMVM-S 4	1.04	0.05	0.80	0.64				
SMVM-S 5	1.01	0.05	0.79	0.62				
SMVM-S 6	1.00	0.00	0.61	0.37	Cognitive liveliness	0.76	0.83	0.62
SMVM-S 7	1.72	0.11	0.96	0.91				
SMVM-S 8	1.51	0.09	0.76	0.58				
SMVM-S 9	1.00	0.00	0.61	0.37	Emotional energy	0.75	0.81	0.52
SMVM-S 10	1.18	0.08	0.73	0.54				
SMVM-S 11	1.42	0.09	0.83	0.69				
SMVM-S 12	1.09	0.08	0.68	0.47				

^a^ Unstandardized coefficient. ^b^ Standard error. ^c^ Standardized coefficient. ^d^ Squared multiple correlation. ^e^ Raykov composite reliability for correlated errors. ^f^ Composite reliability. ^g^ Average variance extracted Note. All factor loads were significant.

**Table 3 ijerph-17-09590-t003:** Pearson correlation coefficients.

	1	2	3	4	5	6	7	8
Total physical activity (1)	-							
Physical strength (2)	0.22 **	-						
Cognitive liveliness (3)	0.04	0.39 **	-					
Emotional energy (4)	0.04	0.38 **	0.29 **	-				
Satisfaction with life (5)	0.06	0.42 **	0.33 **	0.33 **	-			
Dysphoria (6)	−0.05	−0.22 **	−0.21 **	0.08	−0.28 **	-		
Social dysfunction (7)	−0.11	−0.36 **	−0.29 **	−0.15 *	−0.40 **	0.23 **	-	
Loss of confidence (8)	−0.04	−0.25 **	−0.29 **	−0.09	−0.33 **	0.57 **	0.31 **	-
Academic performance (9)	0.06	0.16 **	0.15 *	0.12 *	0.15 **	−0.14 *	−0.03	−0.06

* *p* < 0.05, ** *p* < 0.01.

**Table 4 ijerph-17-09590-t004:** Goodness of fit indices for the non-nested models.

	χ^2^ S-B	df	NNFI	CFI	SRMR	RMSEA	90% CI of RMSEA	CAIC ^e^
Model 1 ^a^	106.03 ***	59	0.89	0.92	0.05	0.05	(0.04–0.07)	−291.236
Model 2 ^b^	108.82 ***	62	0.90	0.92	0.05	0.05	(0.03–0.06)	−308.643
Model 3 ^c^	114.81 ***	60	0.87	0.91	0.06	0.05	(0.04−0.07)	−289.193
Model 4 ^d^	278.63 ***	62	0.52	0.62	0.18	0.11	(0.09−0.12)	−138.838

^a^ Saturated model. ^b^ No direct effect. ^c^ No effect from predictor variable to the mediator. ^d^ No effect from the mediator to outcome. ^e^ Consistent Akaike information criterion *** *p* < 0.001.

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
