# Peer review of "The Shirom-Melamed Vigor Measure for Students: Factorial Analysis and Construct Validity in Spanish Undergraduate University Students"

_ijerph, 2020, doi:10.3390/ijerph17249590_

Round 1

Reviewer 1 Report

Dear Authors,

Overall, a very valuable article has been produced that contains scholarly values of outstanding importance to this topic.

The aim of the research is well-organized and the authors present the given topic in ideal proportions in the introduction.

The manuscript contains the ideal number of literature needed to get to know the topic.

Each of the two parts studies presented in the manuscript is scientifically relevant. The methodology of the studies was analyzed in enough detail by the authors. The comparison of the scales used is relevant and the conduct of the confirmatory factor analysis is scientifically well-organized.

In the discussion section, the authors refer well to the results in the literature and properly analyze their own results. They also effectively highlight potential limitations with their research.

The only lack or problem of this article is in the conclusion chapter, as this in itself is difficult to hold. It would be worth synthesizing this with the previous chapter and drawing the appropriate conclusions. It would be worthwhile to describe in a few lines how the results may correspond to the practice. So how can the organizations or other actors use these results? Furthermore, it could be further specified what additional scales could be used to compare the results obtained.

Overall, it would be important to highlight the further practical applicability of particularly valuable and validated results in the last part of the manuscript.

Author Response

Comment #1

Overall, a very valuable article has been produced that contains scholarly values of outstanding importance to this topic.

The aim of the research is well-organized and the authors present the given topic in ideal proportions in the introduction.

The manuscript contains the ideal number of literature needed to get to know the topic.

Each of the two parts studies presented in the manuscript is scientifically relevant. The methodology of the studies was analyzed in enough detail by the authors. The comparison of the scales used is relevant and the conduct of the confirmatory factor analysis is scientifically well-organized.

In the discussion section, the authors refer well to the results in the literature and properly analyze their own results. They also effectively highlight potential limitations with their research.

Authors’ response #1

We greatly appreciate your words. This is an important encouragement to our work

Comment #2

The only lack or problem of this article is in the conclusion chapter, as this in itself is difficult to hold. It would be worth synthesizing this with the previous chapter and drawing the appropriate conclusions.

Authors’ response #2

The journal's norms indicate the need to include a section of conclusions. However, we consider your suggestions about the content and we address them indicating where these changes are made in the following responses.

Comment #3

It would be worthwhile to describe in a few lines how the results may correspond to the practice. So how can the organizations or other actors use these results? Furthermore, it could be further specified what additional scales could be used to compare the results obtained.

Authors’ response #3

We greatly appreciate this suggestion and, accordingly, we have included applied issues about the usefulness of our results in university contexts and the use of additional variables to improve their applicability (p. 8, lines 319-331).

Comment #4

Overall, it would be important to highlight the further practical applicability of particularly valuable and validated results in the last part of the manuscript.

Authors’ response #4

Following your suggestion and also responding your comment #3, those issues that you mentioned have already been included in the document.

Reviewer 2 Report

The paper addresses a relatively important question in the area of university context, namely, the factorial analysis and construct validity of a Spanish validation of Shirom-Melamed Vigor Measure applied to Spanish university context. Thus, the authors attempt to address this question with a questionnaire-based study that examines a mediator model role of academic vigor in the physical activity-well-being/educational outcomes link.  Results that the Spanish validation scale provided a three-factorial structure, and also confirmed that academic vigor might be a potential mediator in the link among physical activity and well-being and academic outcomes. There were however several concerns that would need to be addressed to support the publication of this manuscript.

I believe that studying the relations between PA, vigor and academic/well-being outcomes in university context is intriguing. In particular, I appreciated working with two different undergraduates. However, the study is limited using a self-report cross-section design to examine mediating factors which should be underlined in the limitation sections.kld

Title

The title of the article is quite long, 5 lines! I suggest to reduce it, using the most important of the two parts. One suggestion: Factorial analysis, construct validity and mediating role of Shirom-Melamed vigor measure in Spanish undergraduate students.

Introduction

The authors suggest the potential mediating role of vigor in the link between PA and academic/well-being indicators among undergraduate students. From a conceptual point of view, the authors highlight that vigor might be an underlying mechanism in that link. However, no theoretical or empirical justifications for this potential link are offered. To support their mediational hypothesis, the authors should offer some theoretical and empirical reasons that justify how vigor could act as such a mediator in such relationship.

Method

Please report whether the research followed the guidelines of the university's ethics commission, especially when requesting confidential data such as academic performance.

With regard to academic performance, it is unclear whether the researchers collected the grades from the university school administration for all undergraduate students or were self-reported by undergraduates. In that case, it should be clear that it is an indicator of “self-report academic performance” and its consequent inflation and social desirability biases in the study limitations. If I have understood well the academic performance collection procedure, the grades data with freshmen students were reported earlier (access to university exam) while collection of their self-reported physical activity and other mental health variables were collected once they were in university course, so they were collected later. These data should be taken with great caution since, according to their study 2, physical activity influences academic performance and not vice versa. One option is removing these freshman data from final analysis or highlight in limitation to take into consideration

Discussion

The PA model explained 4% of the variance in the levels of vigor, do author have some explanations for that low percentage of variance? It does not seem that it is such a key mediator of the PA or, at least, is tentative to think that other unexamined variables have a greater explanatory weight in the academic vigor levels of undergraduate’s students. Similarly, its potential mediating role has more to do with variables of well-being and mental health than with academic performance (71% of variance in mental health vs. 6% of self-reported academic performance). Can the authors discuss some plausible explanations for such findings? These findings seem that academic vigor plays a more fundamental role on mental health indicators than on academic performance. Some discussion or practical implication on these results would be welcome in the discussion Also, according to your results, it is possible that the proposed relationship might be examined as a sequential mediation model, including the two-mediator chain, such as AP-Vigor-Mental Health-PA. Future studies should examine this possibility in deeper details using prospective design.  In the conclusions, the authors suggest that there are some dimensions of vigor (physical aspects of vigor) quite contrasted. Despite obtaining a Spanish adaptation of SMVM-S with a factorial structure with 3 subscales, they have not tested which dimension of the SMVM-S could be more associated with the different DVs. Of course, it could go beyond the object of the present research, however, at least one correlation table would help to know the significant relationships between subscales and different DVs and how robust the associations are. Also, future studies might use multiple mediation to test the potential specific indirect effects of the different SMVM-S dimensions and whether they all exert the same influence on DVs and compare indirect effect between subscales to analyze if some vigor subscales exert a significant higher influence on different indicators of well-being and academic performance.

The authors emphasize in the limitations the temporal importance of the moment of physical activity. However, it is equally important to propose longitudinal approaches, especially when mediation analyzes are carried out where causal relationships are established between PA academic performance and mental health. It would be necessary to underline this issue in the limitations and future research.

In conclusion, the originality of the topic is also an outstanding point of this paper. In my opinion, its major contribution is to provide a new Spanish validation of SMVM-S in the university context and to examine the potential role of vigor between PA and mental health through two different studies. There are no remarkable faults in the statistical treatment or its methodology, but I suggest some correlation table to better understand the relations between variables and the inclusion of some limitations and practical and theoretical implications that deserve further explanation. In the same way, the clarity of presentation is adequate and acceptable.

Apart from that, I consider that the paper is acceptable for publication in the journal if the authors solve the concerns listed above. I hope that these comments, suggestions, will be instrumental in helping you do so.

Thank you for the opportunity to review this manuscript; I hope that this feedback is useful to help the authors revise their work.

Author Response

Comment #1

I believe that studying the relations between PA, vigor and academic/well-being outcomes in university context is intriguing. In particular, I appreciated working with two different undergraduates. However, the study is limited using a self-report cross-section design to examine mediating factors which should be underlined in the limitation sections.kld

Authors’ response #1

We appreciate your valuation of the use of different samples and, according to your suggestion, we have added as a limitation, within the discussion (pp. 8-9, lines 332-336) the type of design used.

Title

Comment #2

The title of the article is quite long, 5 lines! I suggest to reduce it, using the most important of the two parts. One suggestion: Factorial analysis, construct validity and mediating role of Shirom-Melamed vigor measure in Spanish undergraduate students.

Authors’ response #2

Taking into account your suggestion, we have finally opted for a title very similar to the one you propose: “The Shirom-Melamed Vigor Measure for students: Factorial analysis and construct validity in Spanish undergraduate university students” (p. 1, lines 2-4)

Introduction

Comment #3

The authors suggest the potential mediating role of vigor in the link between PA and academic/well-being indicators among undergraduate students. From a conceptual point of view, the authors highlight that vigor might be an underlying mechanism in that link. However, no theoretical or empirical justifications for this potential link are offered. To support their mediational hypothesis, the authors should offer some theoretical and empirical reasons that justify how vigor could act as such a mediator in such relationship.

Authors’ response #3

We have added theoretical arguments to help understand the mediating role that vigor may play in the relationship between BP and different outcomes. Despite this is the first study in educational contexts that provides empirical evidence on these relationships, we do include some studies developed in work contexts (p. 2, lines 65-72).

Method

Comment #4

Please report whether the research followed the guidelines of the university's ethics commission, especially when requesting confidential data such as academic performance.

Authors’ response #4

The protocol was approved by the Ethics Committee of the University of Jaén. We have now included this information and the code number in the paper (p. 3, lines 95-96; p. 5, lines 166-167).

Comment #5

With regard to academic performance, it is unclear whether the researchers collected the grades from the university school administration for all undergraduate students or were self-reported by undergraduates. In that case, it should be clear that it is an indicator of “self-report academic performance” and its consequent inflation and social desirability biases in the study limitations.

Authors’ response #5

Considering these possible biases, we decided to use a GPA self-report measure based on other studies (see reference 1 below) that have found very high correlations between both types of measures

Comment #6

If I have understood well the academic performance collection procedure, the grades data with freshmen students were reported earlier (access to university exam) while collection of their self-reported physical activity and other mental health variables were collected once they were in university course, so they were collected later. These data should be taken with great caution since, according to their study 2, physical activity influences academic performance and not vice versa. One option is removing these freshman data from final analysis or highlight in limitation to take into consideration

Authors’ response #6

Taking into account your appropiated comment, we have carried out sensitivity analyzes, leaving out those freshmen students. No significant changes in the results are found. Only some global adjustment indices of the model reduce their value by a few tenths but without affecting the interpretation of the results. We indicate this point in the discussion section (p. 9, lines 345-351)

 Discussion

Comment #7

The PA model explained 4% of the variance in the levels of vigor, do author have some explanations for that low percentage of variance? It does not seem that it is such a key mediator of the PA or, at least, is tentative to think that other unexamined variables have a greater explanatory weight in the academic vigor levels of undergraduate’s students.

Authors’ response #7

As you correctly indicate, and also as reflected in the heuristic model of vigor at work [see reference 2 below], vigor is not only the result of internal resources such as physical activity. External resources related to organizational, group and task factors (e.g. levels of autonomy, leadership style developed by teachers, planning and design of tasks to be carried out by students ...) would also be responsible for the levels of affect experienced by the students and should obviously be considered in future research. We have included it in the discussion section (p. 8, lines 296-301).

Comment #8

Similarly, its potential mediating role has more to do with variables of well-being and mental health than with academic performance (71% of variance in mental health vs. 6% of self-reported academic performance). Can the authors discuss some plausible explanations for such findings? These findings seem that academic vigor plays a more fundamental role on mental health indicators than on academic performance. Some discussion or practical implication on these results would be welcome in the discussion

Authors’ response #8

The psychological correlates of GPA are varied, including intellective factors, but also motivational and self-regulatory process, personality traits or environmental factors [see reference 3 below]. Taking into account that in our study we have not considered these variables, this could explain the percentage of variance obtained in the explanation of academic performance. We note this extreme in the discussion (p. 8, lines 302-305)

Comment #9

Also, according to your results, it is possible that the proposed relationship might be examined as a sequential mediation model, including the two-mediator chain, such as AP-Vigor-Mental Health-PA. Future studies should examine this possibility in deeper details using prospective design.

Authors’ response #9

We fully agree that testing these models with multiple mediators would help to differentiate between the role of variables that we now consider, as a whole, as result or dependent variables. It would also help to carry out an evaluation of our variables at various moments in time throughout the course (longitudinal design) since this could help to understand the meaning of the relationships raised in our model. We make a reference in that sense (pp. 8-9, lines 332-336)

Comment #10

In the conclusions, the authors suggest that there are some dimensions of vigor (physical aspects of vigor) quite contrasted. Despite obtaining a Spanish adaptation of SMVM-S with a factorial structure with 3 subscales, they have not tested which dimension of the SMVM-S could be more associated with the different DVs. Of course, it could go beyond the object of the present research, however, at least one correlation table would help to know the significant relationships between subscales and different DVs and how robust the associations are. Also, future studies might use multiple mediation to test the potential specific indirect effects of the different SMVM-S dimensions and whether they all exert the same influence on DVs and compare indirect effect between subscales to analyze if some vigor subscales exert a significant higher influence on different indicators of well-being and academic performance.

Authors’ response #10

Following this recommendation, we have included a table with the results of Pearson's correlation coefficients (Table 3. pp. 6-7, lines 261-263) and their associated significance levels. Indeed, the differential influence of the vigor dimensions on the dependent variables transcends the objectives of this study, but we do believe that the table that has been suggested to us to include, helps to understand these potential differences.

Comment #11

The authors emphasize in the limitations the temporal importance of the moment of physical activity. However, it is equally important to propose longitudinal approaches, especially when mediation analyzes are carried out where causal relationships are established between PA academic performance and mental health. It would be necessary to underline this issue in the limitations and future research.

Authors’ response #11

As we have indicated in a previous response, we have included, as a possibility for future research, the use of this type of design you suggest.

Comment #12

In conclusion, the originality of the topic is also an outstanding point of this paper. In my opinion, its major contribution is to provide a new Spanish validation of SMVM-S in the university context and to examine the potential role of vigor between PA and mental health through two different studies. There are no remarkable faults in the statistical treatment or its methodology, but I suggest some correlation table to better understand the relations between variables and the inclusion of some limitations and practical and theoretical implications that deserve further explanation. In the same way, the clarity of presentation is adequate and acceptable.

Authors’ response #12

We appreciate the comments and, as mentioned before, we have included the table with the results of the correlation analysis as suggested.

Comment #13

Apart from that, I consider that the paper is acceptable for publication in the journal if the authors solve the concerns listed above. I hope that these comments, suggestions, will be instrumental in helping you do so.

Authors’ response #13

We greatly appreciate your feedback and we are totally convinced that your suggestions have contributed substantially to the improvement of this paper.

References:

  1. Cassady, J. C. (2000). Self-Reported GPA and SAT: A Methodological Note. Practical Assessment, Research, and Evaluation, 7, article 12. https://doi.org/10.7275/5hym-y754
  2. Shirom, A. (2011). Vigor as a positive affect at work: Conceptualizing vigor, its relations with related constructs, and its antecedents and consequences. Review of General Psychology, 15(1), 50–64. https://doi.org/10.1037/a0021853
  3. Richardson, M., Abraham, C., & Bond, R. (2012). Psychological correlates of university students' academic performance: a systematic review and meta-analysis. Psychological Bulletin, 138(2), 353–387. https://doi.org/10.1037/a0026838

Reviewer 3 Report

The effect of Physical Activity on Vigor and its redundancy in other variables such as Mental Health, Satisfaction with Life and Academic Performance is of relevant interest, which has been analyzed with the university population. The manuscript provides the need for more depth in those variables that can be key in the development of a healthier lifestyle, where a minimum of weekly hours of physical activity are dedicated and avoid unhealthy behaviors in terms of diet and sedentary lifestyle , which have had a relevant growth in the university population in recent years and that could increase, even more, due to the effects of the COVID-19 pandemic. Therefore, analyzing the relationship of variables such as Vigor on Physical Activity and its benefits will be a promoter in the creation of work models that can be implemented, both in the university population and in primary and secondary school students, in order to promote values, attitudes and behaviors that lead to a healthy lifestyle.

To improve the manuscript it is recommended:

- Shorten the title, it is too long.

-Expand the theoretical introduction, carrying out a broader and more detailed explanation of the benefits of physical activity on mental health, lifestyle and academic performance in different populations, as well as better clarify the concept of the variable Vigor and its relationship with the variables mentioned and which are analyzed in this work.

-Make a better definition of the objectives to clarify the purpose of the study, as well as its contribution.

-Expand the conclusions section and connect them with the objectives once they are formulated again.

-Expand the applicability and usefulness of the variables analyzed, as well as the future lines of intervention and prevention that could be derived from this work.

-Include bibliographic references from this year.

The work is very well structured and well founded.

Author Response

Comment #1

- Shorten the title, it is too long.

Authors’ response #1

Taking into account your suggestion, we have finally opted for: “The Shirom-Melamed Vigor Measure for students: Factorial analysis and construct validity in Spanish undergraduate university students” (p. 1, lines 2-4)

Comment #2

-Expand the theoretical introduction, carrying out a broader and more detailed explanation of the benefits of physical activity on mental health, lifestyle and academic performance in different populations, as well as better clarify the concept of the variable Vigor and its relationship with the variables mentioned and which are analyzed in this work.

Authors’ response #2

Following this indication, in the introduction section, we have added additional theoretical arguments and empirical studies that would help the reader understand the relationships that vigor maintains with the rest of the variables analyzed in our study and the possible explanatory mechanisms that deal with these relationships (p. 2, lines 64-72). Regarding the benefits of PA, extending the explanation goes beyond the objectives of this study and with the references we include, readers can expand the information on this aspect if they wish.

Comment #3

-Make a better definition of the objectives to clarify the purpose of the study, as well as its contribution.

Authors’ response #3

We have added a small additional paragraph to each objective, thus, the purpose of the studies can be better understood (p. 2, lines 82-87).

Comment #4

-Expand the conclusions section and connect them with the objectives once they are formulated again.

Authors’ response #4

We have connected, within the conclusion section, our results with the starting objectives of both studies (p. 9, lines 359-363)

Comment #5

-Expand the applicability and usefulness of the variables analyzed, as well as the future lines of intervention and prevention that could be derived from this work.

Authors’ response #5

We have added some practical implications derived from our results and also raised future options that open up in terms of intervention and prevention (p. 8, lines 319-331).

Comment #6

-Include bibliographic references from this year.

Authors’ response #6

In the list of references, you can check that we have followed your suggestion.

Comment #7

The work is very well structured and well founded.

Authors’ response #7

We greatly appreciate your comment.